

# Impact of Landes forest fires on air quality in France during the summer 2022

Laurent MENUT[1], Arineh CHOLAKIAN[1], Guillaume SIOUR[2], Rémy LAPERE[3], Romain PENNEL[1], Sylvain MAILLER[1], and Bertrand BESSAGNET[4]

[1]Laboratoire de Météorologie Dynamique (LMD), Ecole Polytechnique, IPSL Research University, Ecole Normale Supérieure, Université Paris-Saclay, Sorbonne Universités, UPMC Université Paris 06, CNRS, Route de Saclay, 91128 Palaiseau, France
[2]Univ Paris Est Créteil and Université Paris Cité, CNRS, LISA, F-94010 Créteil, France
[3]Université Grenoble Alpes, CNRS, IRD, Grenoble INP, IGE, 38000 Grenoble, France
[4]European Commission, Joint Research Centre (JRC), 21027 Ispra, Italy

**Correspondence:** Laurent Menut, menut@lmd.ipsl.fr

**Abstract.** The atypical huge forest fires observed in France during the summer of 2022 are modelled using the CHIMERE model. Scenario simulations are performed without and with these fires to quantify the impact of these extra emissions on air quality thresholds exceedances. Additional processes are added in the model to better simulate fire emissions and then have more realistic simulations. The fires influence the characteristics of the surface by destroying the vegetation and creating new erodible surfaces. This increases the mineral dust emissions, but also reduces the Leaf Area Index, then decreases the biogenic emissions and the dry deposition of gases such as ozone. Results show that the fires induce numerous increases in surface ozone and Particulate Matter concentrations close to the sources but also in downwind remote sites such as the Paris area. During the period of the most intense fires in July, the impact of concentrations is mainly due to emissions themselves, when later, in August, ozone and PM concentrations continue to increase but this time due to changes in the burnt surfaces.

## 1 Introduction

Forest fires are frequent in summer and in Europe, mainly in the south. They are usually observed in Greece, Spain, Portugal and can last several weeks, (San-Miguel-Ayanz et al., 2022). In addition to the destruction of burning vegetation, these fires emit numerous air pollutants that can degrade the air quality in the areas downwind of the smoke plumes. In France, these fires are more rare. But during the summer 2022, numerous huge and atypical forest fires were observed. The Landes de Gascogne forest in south-western France covers an area of about 1500000 ha, mostly belonging to the departments of Gironde (to the North) and Landes (to the South). Mostly planted during the 19[th] century, this forest is primarily composed of maritime pine (Mora et al., 2014). Major episodes of wildfires occurred in this large forest in the past, the most dramatic being the megafire of 1949, which burnt 50000 ha in the Gironde department and caused 82 deaths. Recent significant events occurred in August 2015 in the vicinity of Bordeaux (500 ha burnt) and April 2017 (1100 ha burnt). With an increased urbanization and demographic growth in this area, the risk associated to wildfires mechanically increases. Protection and management measures against wildfires have been taken in this forest after the 1949 disaster. These measures rely on a strong implication of the local



economic actors who grow and harvest the forest. However, this implication has been weakened in the recent years due to many factors including economic hardship for the forest sector following the destructions caused by storms Martin (1999) and Klaus (2009). Therefore, the management of the forest by economic actors is not as intense as it used to be, easing the propagation of wildfires, while climate change favors an increased risk of wildfires, (Huang et al., 2015).

The 2022 fire season was the worst in this region since 1949. Three main episodes have occurred during this summer season, first from July 12 to July 22, with two major fire events close to Landiras (13800 ha burned as of July 20) and La-Teste-de-Buch (7000 ha burned as of July 20). However, the Landiras fire, still propagating underground due to the presence of peat, began active again on 9 August, burning another 7400 ha of forest between 9 and 14 August and a last one in September burning 3400 ha in Saumos. These events destroyed more than 30000 ha of forest in Gironde during the 2022 fire season. One explanation is the unusual heat wave observed in France during this summer: the forest and its soil are dryer than usual and high wind speed were observed. In addition, the region experienced very low precipitations compared to seasonal norms, (Toreti et al., 2022).

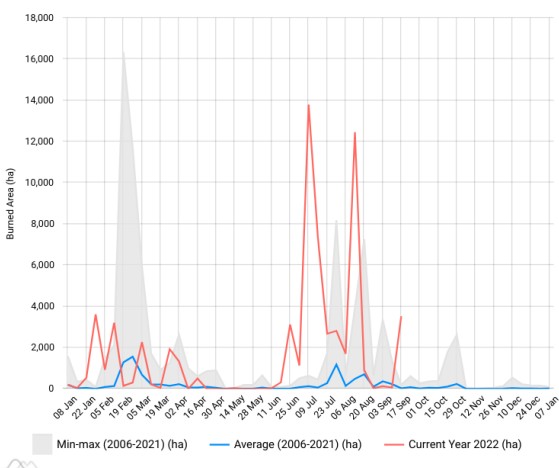

**Figure 1.** *Seasonal trend of weekly burned areas (ha) in France as an average over the period 2006-2021 and for the year 2022 (until 13 September). The figure is extracted from the EFFIS database (https://effis.jrc.ec.europa.eu/apps/effis.statistics/estimates).*

Seasonal trend of vegetation fires is presented in Figure 1. The figure is extracted from the European Forest Fire Information System (EFFIS). The blue curve shows the weekly average burned area (ha) in France for the period ranging from 2006 to 2021. The maximum of burned area are February, July and August and do not exceed 2000 ha. The grey shade shows the maximum values recorded during this period and values may reach 18000 ha in February. The red curve presents the burnt area for the year 2022 only. Summer 2022 is characterized by two extreme peaks in weekly burnt areas, with 14000 ha burnt in one week in July and 13000 ha in one week of August, more than any other sole summer week in the 2006-2021 period.

By emitting gas and particles in the atmosphere, vegetation fires change directly the atmospheric composition downwind the fire plume, (Jaffe and Wigder, 2012), (Rea et al., 2015). It has a direct impact on surface concentrations of ozone, nitrogen



oxides and particulate matter, then on possible pollution peaks monitored by air quality networks. Some other impacts of fires exist: they have a direct effect of aerosol on meteorology by attenuating the solar radiation, (Reid et al., 2005), and changing microphysics, (Grell et al., 2011). They also have an impact on other natural emissions such as mineral dust, (Wagner et al., 2018), (Menut et al., 2022b). A possible impact exists also on biogenic emissions, the fires destroying the vegetation potentially emitting chemical species, (Vieira et al., 2023).

The main questions addressed in this study are: (i) Is the model able to simulate the fires plumes? (ii) Do the biomass burning have a significant impact on mineral dust emissions, dry deposition or biogenic emissions by changing the surface? (iii) Are the fires plumes responsible of additional pollution peaks in urbanized areas? To answer these questions, several simulations are performed with regional modelling and comparisons to observations. In Section 2, the CHIMERE model used is presented as well as its specific configurations and the model developments made for this study. In Section 3, 4 and 5, the results of the simulations are presented.

## 2   The modeling system

The modeling system is presented in this section with the models used, the databases employed as forcings and the main changes made in the last models versions.

### 2.1   The models set-ups

In this study, we use the CHIMERE model v2020r3, (Menut et al., 2021) forced by IFS ECMWF meteorological fields, (Haiden et al., 2022). Two model domains are defined, one with an horizontal resolution of 50 km, the second one, nested inside the largest one, with an horizontal resolution of 15 km (Figure 2). The larger domain is designed to have the boundary conditions far from France where fires are studied and also to have an explicit representation of the numerous natural emissions around and in France (mineral dust in Africa, sea salt, biogenic emissions). Figure B1 presents the domain border in red as well as the Leaf Area Index (LAI), in $m^2.m^{-2}$, for the month of August. The second domain is centered over France and is dedicated to have a good resolution to well capture the thin plumes generated by forest fires. The two domains are presented in Figure 2. Note that all results will be presented using the simulation made with the inner domain, with a resolution of 15 km. CHIMERE has 15 vertical levels from the surface to 300 hPa.

Several tens of chemical species, gas and aerosol, are modelled. For aerosols, ten bins are used from 0.01 to 40 $\mu m$. Emissions include several contributions such as anthropogenic, biogenic, sea-salt, dimethylsulfide, biomass burning, lightning $NO_x$ and mineral dust. The anthropogenic emissions are those of CAMS, (Granier et al., 2019). Having no available data for the summer 2022, we used the 2018 year for these emissions. Indeed, we avoided the years 2019, 2020 and 2021 to avoid lockdown effects or other perturbations due to this very particular COVID19 period (Menut et al., 2020). The dry deposition is modelled following the Zhang et al. (2001) scheme and the wet deposition following Wang et al. (2014).

The biomass burning emissions are those of CAMS as described in (Kaiser et al., 2012) and presented in Figure 3 for the modelled domain with 15 km resolution. These surface fluxes are vertically redistributed as described in Menut et al. (2018).





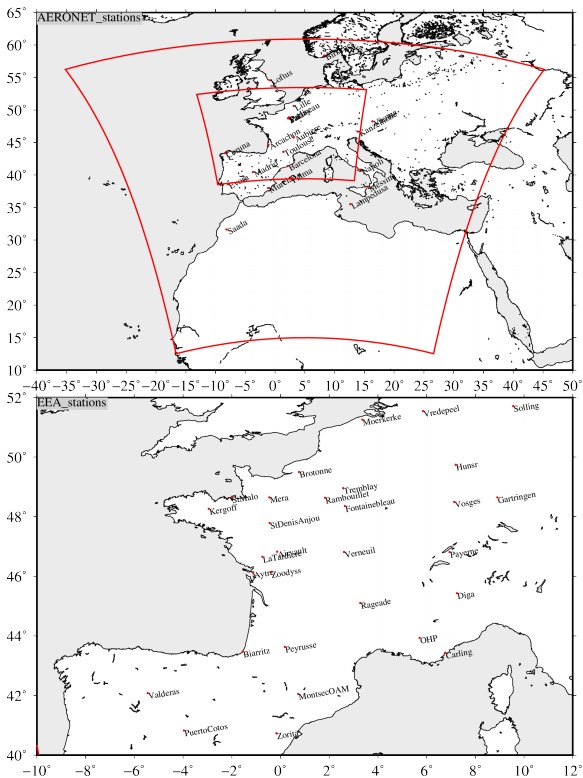

**Figure 2.** *Maps of measurements stations of EEA and AERONET. The two nested model domains are represented as red frames. The largest one has an horizontal resolution of 50 km, when the second one has an horizontal resolution of 15 km.*

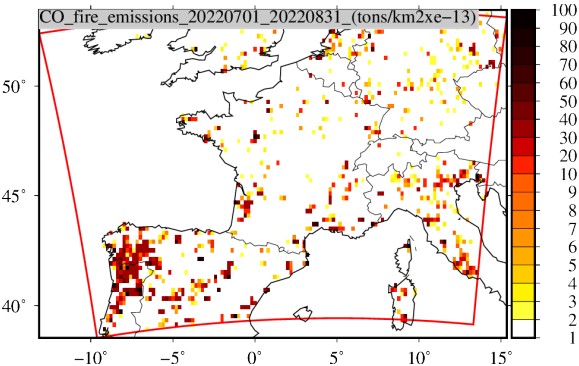

**Figure 3.** *Time-averaged surface flux of CO emitted by fires during the months of July and August 2022 and calculated using the CAMS fires product.*





Mineral dust emissions are calculated using the Alfaro and Gomes (2001) scheme with the numerical optimization presented
in Menut et al. (2005). Sea-salt emission is calculated using the Monahan (1986) scheme. $NO_x$ by lightning are calculated
following Menut et al. (2020) using the Price and Rind (1993) parameterization. Primary Particulate Matter (PPM) can also be
emitted by resuspension process as described in Vautard et al. (2005). The biogenic emissions are modelled using the MEGAN
model, (Guenther et al., 2012), with the Leaf Area Index with 30 seconds resolution and 8-days frequency, (Sindelarova et al.,
2014).

## 2.2 Impact of fires on other natural emissions

The fires destroy the environment and thus have an impact on potential natural emissions and processes. In this study, three
different impacts of vegetation fires are studied. First, a non-negligible impact is on the local wind speed and the erodibility. The
local wind speed is enhanced by the massive pyroconvection creating a surface pressure gradient. The burnt surface becomes
more erodible. Depending on the vegetation type, the increase in erodibility can last several months, (Menut et al., 2022b).
The conjunction of an higher wind speed and an higher erodibility leads to higher mineral dust emissions. Another impact is
due to the fires destroying the vegetation then decreasing the Leaf Area Index (LAI). The LAI is involved in the calculation of
two processes in the model. First, the LAI proportionally affects the biogenic emission when less LAI induces less biogenic
emissions. Second, less LAI is also responsible of less dry deposition of gaseous species, having less available leaf surface. To
take into account this effect, the LAI is reduced proportionally to the burnt area (the same percentage of surface) during and
after each fire. Taking these three impacts into account has the effect of increasing dust emissions, reducing biogenic emissions
but also reducing dry deposition.

## 2.3 The simulations

Simulations designed for this study are summarized in Table 1. They all have in common to cover the period from 15 June
to 31 August 2022. The first one is called *nofire* and corresponds to the modelling with all emissions except the forest fires.
The model is used in its *offline* version meaning there are no retroactions from aerosols to meteorology. All other simulations
are with the biomass burning emissions and have a name with "f" for fires. The first simulation with fires is called *f2no* and
corresponds to the emissions of fires but without impact on other processes. It corresponds to the classical use of fires emissions
in chemistry-transport models: only a flux of chemical species is prescribed when a fire is detected. The simulation *f2laibio*
is as *f2no* but with, in addition, the impact of the fires on the LAI used for the calculation of the biogenic emissions. The
simulation *f2laidd* is as *f2no* but with, in addition, the impact of the fires on the LAI used for the dry deposition of gaseous
species. The simulation *f2dust* is as *f2no* but with, in addition, the impact of the fires on the mineral dust emissions as described
in (Menut et al., 2022a). Finally, the simulation *f2all* is the more realistic, taking into account both emissions of the fires and
interactions between fires emissions and surface properties (on LAI for biogenic emissions and dry deposition) and mineral
dust emissions.

The first goal of this study is to have a reference case able to quantify what would have been the atmospheric composition if
the observed fires had not existed. For this question, we will use the *'f2all - nofire'* differences. The second question is to know



| Simulation | Fire emis. | Impact on | | | |
|---|---|---|---|---|---|
| | | LAI | | | Dust emis. |
| | | Bio emis. | Dry Dep. | | |
| nofire | | | | | |
| f2no | ✓ | | | | |
| f2laibio | ✓ | ✓ | | | |
| f2laidd | ✓ | | ✓ | | |
| f2dust | ✓ | | | | ✓ |
| f2all | ✓ | ✓ | ✓ | | ✓ |

**Table 1.** *Simulations performed for this study.*

the impact of the retroactions of fires on dust emissions and the LAI parameter. We will then use in this case the differences between the simulations with impacts against the *f2no* simulation. The analysis of the simulation is performed from 1 July to 31 August 2022. The simulated period from 15 to 30 June considered as a spin-up period is not analyzed.

## 2.4 The observations

Several types of observations are used to quantify the model ability to reproduce these events. First, measurements from surface stations are used. The European Environment Agency (EEA, https://www.eea.europa.eu) provides a full set of hourly data for several pollutants such as particulate matter $PM_{2.5}$ and $PM_{10}$, ozone ($O_3$) and nitrogen dioxide ($NO_2$) for a large number of stations in Western Europe. Only urban, rural and suburban background stations are used, considering that the industrial and traffic ones have an inadequate spatial representativity for model outputs with a spatial resolution of $\Delta x$=15 km. The *AErosol RObotic NETwork* (AERONET, https://aeronet.gsfc.nasa.gov/) level 1.5 measurements are used, (Holben et al., 2001). The AOD at a wavelength of $\lambda$=675 nm is daily averaged and compared to daily averaged modelled values. Maps of the stations for which the measurements were used are presented in Figure 2. The detailed names and location of these stations are provided in Table A1 and Table A2. The map of the AERONET stations shows the entire modelled domain. The second map is a zoom on the region that we will study in more detail. Note that the stations the more close to the studied fires are Airvault (FR09304), LaTardiere (FR23124), Aytre (FR09008) and Zoodyss (FR09302).

Second, and in order to have an information on the vertical, CALIPSO lidar data are used. The CALIOP lidar measurements, on-board the Cloud-Aerosol Lidar Pathfinder Satellite Observation (CALIPSO) satellite (Winker et al., 2010), are analyzed to obtain an aerosol sub-type classification (CALIOP v4.10 product), as proposed in Omar et al. (2010) and Burton et al. (2015). Limitations associated with this aerosol classification are described in Tesche et al. (2013). For the model, a specific





development was performed as described in Menut et al. (2018), using aerosol concentrations to reproduce the categories chosen by the CALIPSO team.

## 3 Impact of fires on aerosol

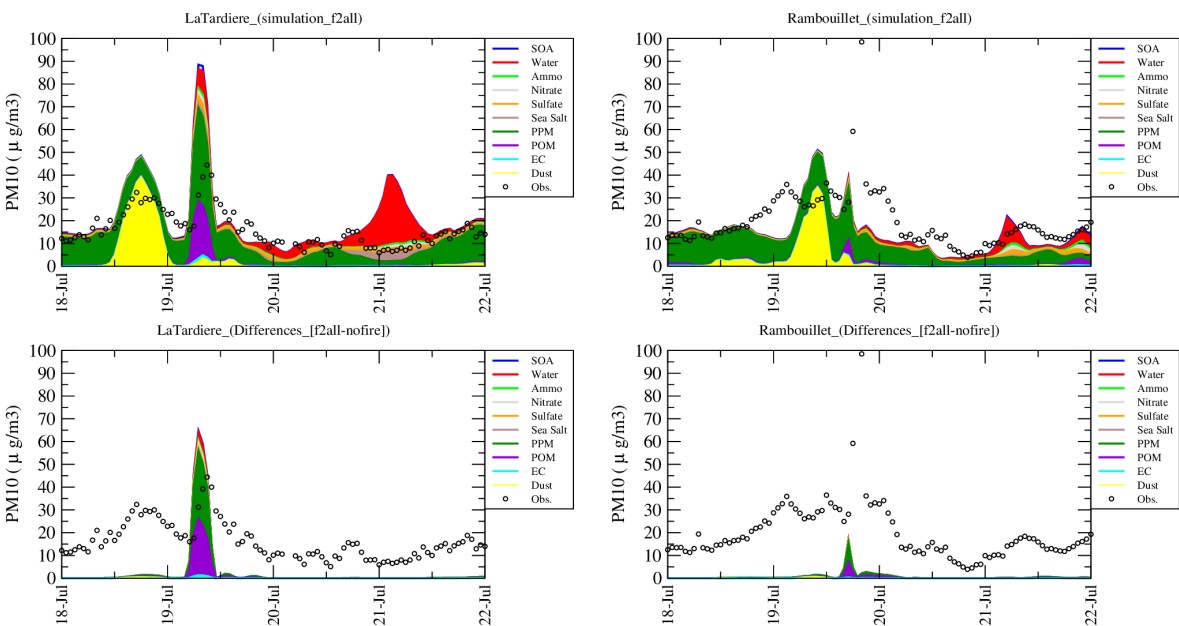

**Figure 4.** *Time-series of absolute values and differences of PM$_{10}$ ($\mu g.m^{-3}$) surface concentrations in LaTardiere and Rambouillet, for the period 18 to 22 July 2022. For the model values, the chemical composition of the PM$_{10}$ is presented.*

In this section, the impact of fires on aerosol is analyzed, first on the aerosols surface concentrations, second on the Aerosol Optical Depth (AOD).

### 3.1 Impact of fires on PM$_{10}$ surface concentrations

The first question is to know if Landes fires have changed the surface concentrations of pollutants close to the source or downwind. Figure 4 presents time-series of PM$_{10}$ hourly surface concentrations (in $\mu g.m^{-3}$). The presented time-period is reduced to 18 to 22 July 2022 in order to have a more precise view of the fire event of 19 July. Time-series are presented for the two sites of LaTardiere (close to the fires) and Rambouillet (in the Paris area) and for the simulation *f2all*. In addition, the time-series presents the modelled chemical composition of the PM$_{10}$. It is not possible for the measurements, providing only the total mass of the aerosol. For the LaTardiere stations, two peaks of PM$_{10}$ are observed and correctly modelled. The first one occurs the 18 July and corresponds to mineral dust. There is also Primary Particle Matter (PPM) concentrations, but they are as a background during the whole period and correspond to resuspension in this agricultural and forest region. The second





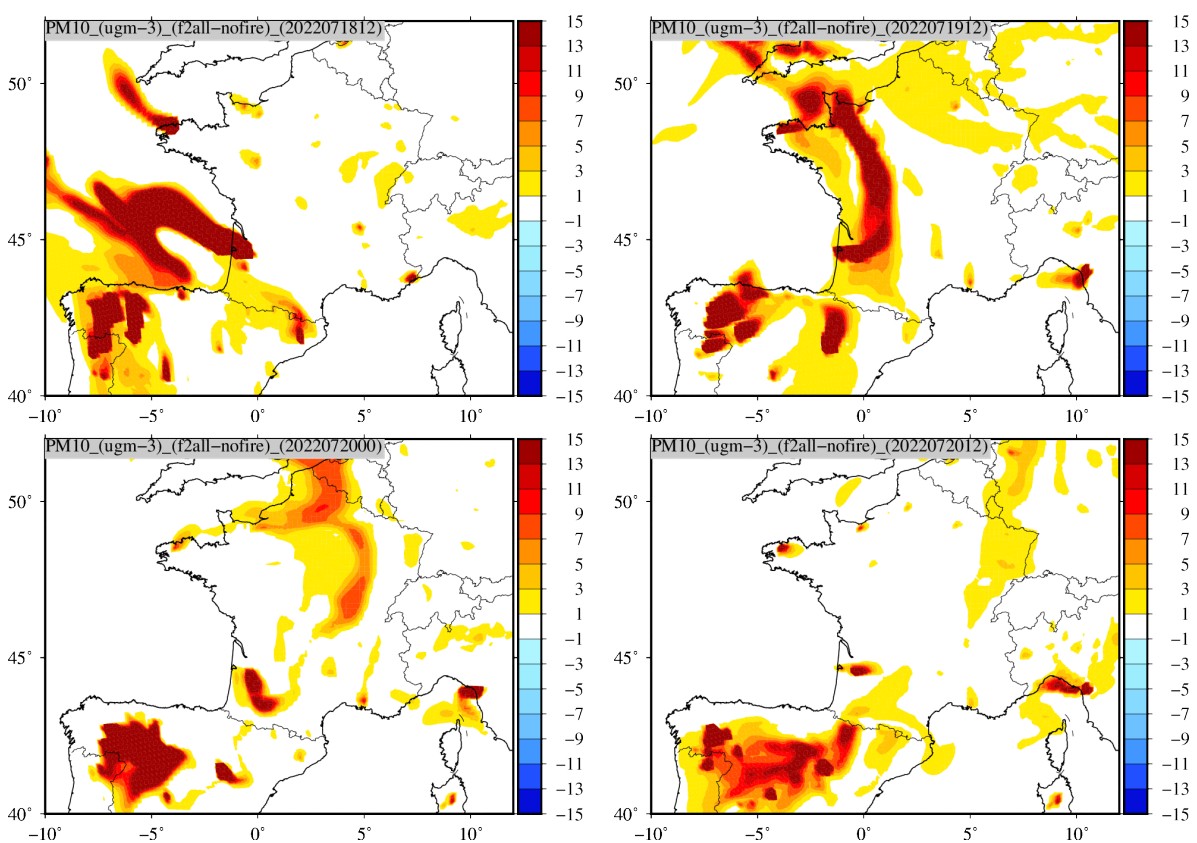

**Figure 5.** *Maps of surface concentrations of $PM_{10}$ ($\mu g.m^{-3}$) for the 18, 19 and 20 July 2022 at 12:00 UTC.*

peak correspond to the forest fires. The model overestimates the measurements but is composed of Primary Organic Matter (POM), signature of the biomass burning. The 21 July at midnight, a peak of water is also modelled, due to a change in the wind direction and air masses coming from the Atlantic sea. For the same site, the time-series of differences between *f2all* and *nofire* are presented. It shows that the only difference between the two simulations occurs the 19 July and is half composed of POM and half composed of PPM. Far from the fires, in Rambouillet, the time-series of *f2all* simulation shows that a similar

peak of dust is modelled, and corresponds to an observed peak. It occurs on 19 July (in place of 18 July in LaTardiere) and corresponds to the transport of the mineral dust plume over France. The second peak, corresponding to the transport of the fires plumes, is underestimated by the model but present and visible in the time-series of differences. When the additional $PM_{10}$ surface concentration due to fires is $\approx 70$ $\mu g.m^{-3}$ in Latardiere, it is only $\approx 10$ $\mu g.m^{-3}$ in Rambouillet.

      In order to have another point of view on $PM_{10}$ surface concentrations, maps are presented in Figure 5. These maps display

the differences between the two simulations *f2all* and *nofire* to spatialize the transport of the biomass burning plumes and to quantify their impact far from the fires areas. The first map represents the 18 July at 12:00 UTC. In France, two main fires are observed: in Landes and in Brittany. The wind has the same direction and the plume is transported toward west over the



Atlantic sea. For this day, there is a priori no impact on land in France. The only impact may be in the South, under the plumes of Portuguese fires. The second map presents the concentrations for the 19 July at 12:00 UTC. The wind has turned and is now from south to north. The fire plume goes towards Brittany, Normandy and Belgium and passes to the west of the Parisian region. The third map is for the 20 July at 00:00 UTC. The plume over France is diluted and is split in two parts: one in the south and one in the north of the Paris area. This explains the underestimation of the model for the stations in the Paris area. The fourth and last map present the concentrations for the 20 July at 12:00 UTC. Differences of surface concentrations are now low, except just over the active fires in Landes and Brittany. High differences are modelled over Spain and Portugal, but impact moderately the surface concentrations in the south of France. Finally, the impact of fires induces positive differences only. The timing of the sources and the transport is realistic. The only lack of the simulations is probably the long range transport of the fire plume, being cut in two and does not pass over the Paris region with high concentrations.

## 3.2 Comparisons to AERONET measurements

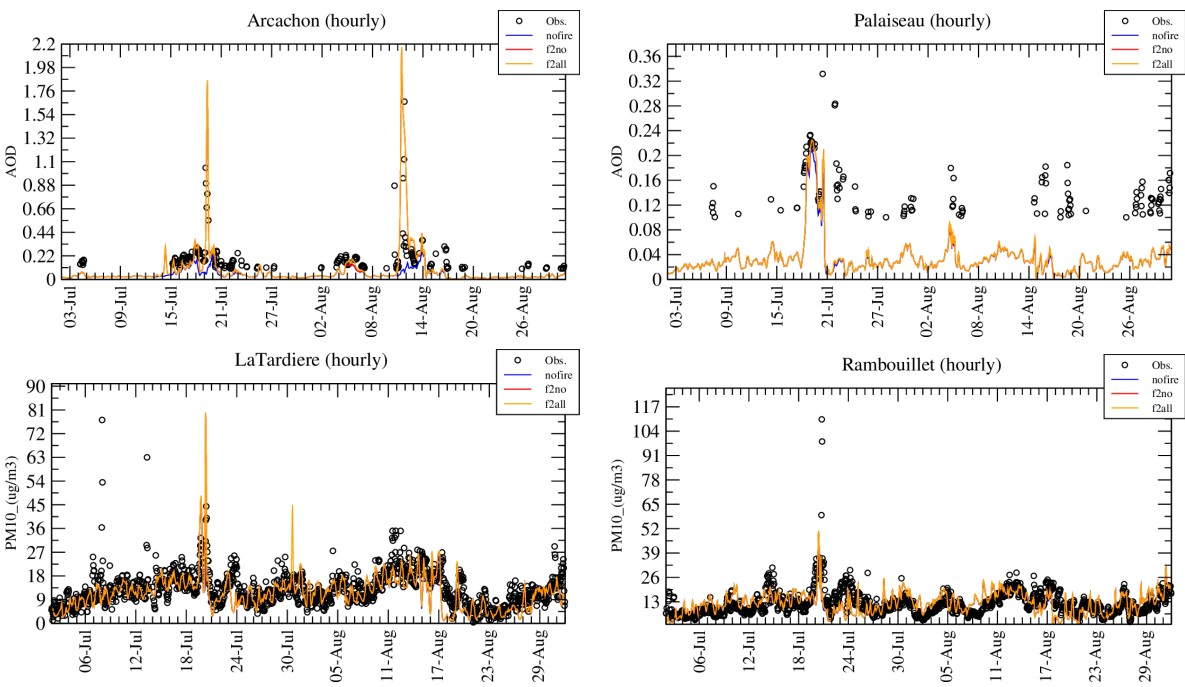

**Figure 6.** *Time-series of (top) hourly Aerosol Optical Depth in Arcachon and Palaiseau and (bottom) PM$_{10}$ surface concentrations in LaTardiere and Rambouillet. The three model simulations nofire, f2no and f2all are compared to the measurements of AERONET (AOD) and EEA (PM$_{10}$).*

Comparison of model concentrations results against measurements are also performed for the Aerosol Optical depth (AOD) using AERONET measurements. Results are presented in Figure 6 for the stations of Arcachon (close to the fires) and Palaiseau (close to Paris). In Arcachon, two important peaks are measured and modelled on 19 July and 10 August. The model is able to





retrieve these peaks at the right time. The differences between the curves show this is only the impact of fires. For Arcachon, we can note that only the first peak is present on the $PM_{10}$ time series in Latardiere (close to Arcachon and the fires). It means that the fires on 19 July are in the boundary layer and impact the surface concentrations, but are probably in altitude on 10

August: they are visible on the AOD time-series but not on surface concentrations. It is true both for the measurements and the model results. In Palaiseau, only the peak of 19 July is visible on the AOD time-series. It is the same for the $PM_{10}$ surface concentrations in Rambouillet (close to Palaiseau). But in this latter case, model values are underestimated: when the model simulates a peak at $\approx 50$ $\mu$g.m$^{-3}$, the measurements show high values $\approx 110$ $\mu$g.m$^{-3}$. It is meaning that the plume coming from Landes reaches the Paris area but is simulated too low compared to the measurements.

In order to refine the analysis on $PM_{10}$ surface concentrations, the modelled aerosol composition is presented in Figure 7 as size distribution. Depending on the data availability, results are presented here for Arcachon and Paris and for the 19 July at 15:00 UTC. Two simulations results are compared to measurements: *nofire* and *f2all*. For the four figures, the same kind of distribution is calculated: two modes are modelled, with a fine mode with a mean mass median diameter $D_p \approx$ 0.1-0.2 $\mu$m and a coarser mode with $D_p \approx$ 1-6 $\mu$m. The fine mode is composed of all kind of modelled aerosols, with a dominant part of PPM,

here due to resuspension. For the coarse mode, the most important contribution of the composition is mineral dust. Note that the Efficient Extinction Section coefficient is superimposed (in dashed line and normalized for the Figure). This coefficient is used to the AOD calculation and it appears that its maximum corresponds to a minimum of concentration in the size distribution: it means that the AOD calculation is very sensitive to the size distribution and the number of bins of the model (even if here it is concentrations at the surface only).

The main difference between *nofire* and *f2all* is for the Arcachon site and the fine mode where a non-negligible contribution of POM is calculated in case of fires. It is the direct impact of biomass burning in the aerosol composition. However, there is no clear differences between the two simulations at the Paris site.

### 3.3 Vertical transport of the fire plume

The differences between the time-series of AOD and surface concentrations of $PM_{10}$ show that the fire plume might have

190 been transported aloft without high concentrations being present at the surface. To verify this hypothesis with the simulations, vertical sections are presented in Figure 8. These cross-sections are presented for the simulation *f2all* and for the difference between the two simulations *f2all-nofire*. The figure presents an iso-longitude cross-section (for longitude -1$^o$, corresponding roughly to the longitude of the Landes fires). The latitude ranges from 40 to 53$^o$N. Two periods are presented: 19 July at 00:00 UTC and 12:00 UTC. At 00:00 UTC, the maximum of $PM_{10}$ are mainly in altitude between 2000 m and 4000 m AGL.

Concentrations close to the surface are low and do not exceed 20 $\mu$g.m$^{-3}$.

Some maximum are modelled in altitude and for latitude between 40 and 43 $^o$N, and between 49 and 53 $^o$N. For the latitude of the Paris area, there is low concentrations of $PM_{10}$ over the whole atmospheric column at 00:00 UTC and close to the surface at 12:00 UTC. The differences between *f2all* and *nofire* show that the most important contribution of fires remain below 5000 m AGL. The maximum of differences are at latitude 44-46 $^o$N at 00:00 UTC and 42 and 50 $^o$N at 12:00 UTC.

There is no important impact modelled for the latitude of the Paris area, $\approx 48$ $^o$N.



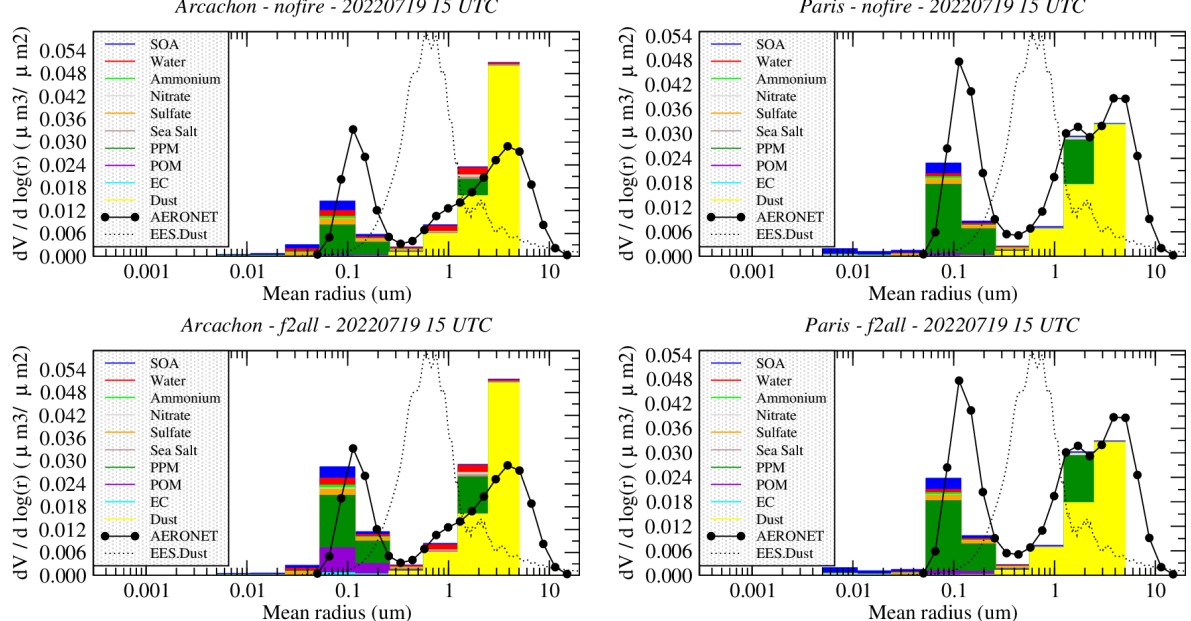

**Figure 7.** *Aerosol size distribution and composition for the 19 July 2022 at 15:00 UTC and for the stations of Arcachon and Paris. Model outputs are compared to the AERONET product. The dashed line represents the Efficient Extinction Section (EES) calculated for mineral dust and normalized to the maximum value of the model for the plot.*

In order to follow the wildfires plumes transported to the north-east, we compare model vertical cross-sections of aerosol concentrations to CALIOP lidar data. The CALIOP lidar is on-board the Cloud-Aerosol Lidar Pathfinder Satellite Observation (CALIPSO) satellite, (Winker et al., 2010). Vertical lidar profiles are analyzed to obtain an aerosol sub-type classification (CALIOP v4.10 product), developed by Omar et al. (2010) and Burton et al. (2015). This classification is built on thresholds of
lidar-derived optical characteristics. Of course, this estimation is uncertain and limitations are quantified in Tesche et al. (2013). For the CHIMERE model results, a specific development was done in Menut et al. (2018) to retrieve the same classification but based on all modelled aerosols. The comparison is presented in Figure 9 for the dataset named *CAL_LID_L2_VFM-ValStage1-V3-41.2022-07-18T13-51-17ZD*. It corresponds to a trajectory quasi-iso-longitude and the data are presented for the latitude from 10 to 60 $^o$N. The CALIOP data are scarce (white areas are for no data) but show that the aerosol plume is mainly between
the surface and 5000 m AGL. It also shows that the aerosol composition is mostly dust and polluted dust. The same type of composition is modelled and analyzed with the model. The locations of the several types of dust is well retrieved by the model, showing that the modelled transport is realistic.





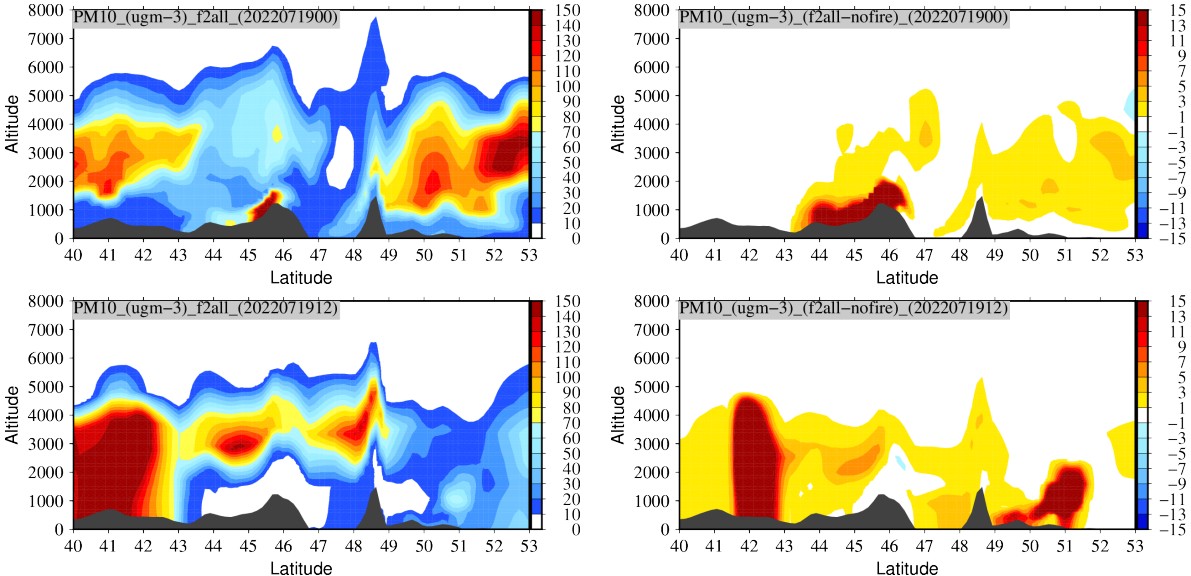

**Figure 8.** *Differences between "f2all" and "nofire" simulations for the PM$_{10}$ concentrations at longitude -1°E and for the 19 July at 00:00 and 12:00 UTC.*

## 4 Impact of fires on surface ozone concentrations

### 4.1 Comparisons between observed and modelled surface concentrations

Time series are presented in Figure 10 for comparison between measured and modelled surface ozone concentrations during the period from 15 July to 15 August 2022. Data are daily averaged frequency in order to highlight the most important differences. The model results are presented for the three simulations *nofire* (no biomass burning emissions), *f2no* (fires but no retroactions on dust and LAI) and *f2all* (fires and retroactions).

For the four stations presented in Figure 10, Biarritz, LaTardiere, Rambouillet and Kergoff, located at various ranges from

the fires (LaTardiere being the closest one), there is no important impact of the fires emissions on daily mean surface ozone concentrations. The concentrations vary a lot from one week to another, but the simulated concentrations are very close to each other. Two periods of higher concentrations are noted both with the model and the measurements: between 12 and 18 July and between 5 and 17 August 2022. These two episodes are observed for the four stations, showing this is a spatially extended episode over the whole France. Values are not very high as daily mean, ranging from 60 to 140 $\mu$g.m$^{-3}$. Results are

also presented as statistical scores in Table 2. These scores are defined in Menut et al. (2019). The best spatial correlation is for the simulation the more realistic *f2all*. But the best temporal correlation is obtained with the *nofire* simulation (R=0.77) even if the two others simulation have very close results (0.77 for *f2no* and 0.76 for *f2all*). The RMSE and the bias are better for *f2all*: with a value of -8.46 $\mu$g.m$^{-3}$, the bias is significantly reduced compared to the two other simulations with values of -10.62 $\mu$g.m$^{-3}$ for *nofire* and -10.08 $\mu$g.m$^{-3}$ for *f2no*.



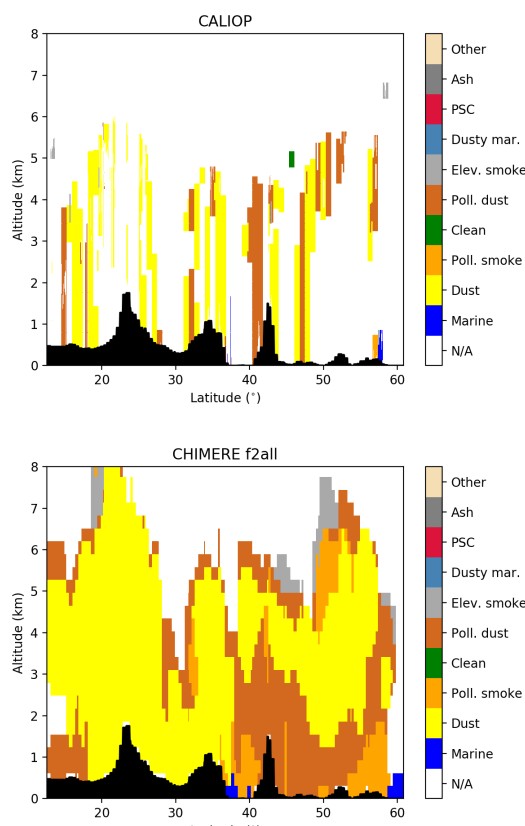

**Figure 9.** *CALIOP and CHIMERE vertical aerosol classification for the 18 July 2022 at 13:51 UTC.*

For the two periods, the type of the differences between the simulations is not the same. During the first period, 12 to 18 July, the simulations with fires (*f2no* and *f2all*) provides ozone concentrations with a peak 10 $\mu$g.m$^{-3}$ higher than the simulation nofire. This is the direct impact of the additional emission due to fires. There is no difference between *f2no* and *f2all*, showing that the secondary effect of fires on mineral dust emissions and LAI have a negligible impact during this period. During the second period with high ozone concentrations, from 5 to 17 August 2022, the three time-series are separated: if *nofire* provides again the lowest ozone concentrations, the *f2all* simulation is now higher than the *f2no* simulation. It means that during this second period the impact of the fires tends to increase the surface ozone production. And this trend increases with time, the surface being modified for the whole on-going simulation.

In order to quantify the differences between the simulations and the observations, Figure 11 presents time-series for the stations of LaTardiere and Rambouillet. The three time-series represent the differences between the observations and the simulations *nofire*, *f2no* and *f2all* for the daily mean values of surface ozone concentrations ($\mu$g.m$^{-3}$). In LaTardiere, the differences are mostly negative: the model has a negative bias compared to observations, underestimating the daily mean



| Simulation | $R_s$ | $R_t$ | RMSE | bias |
|---|---|---|---|---|
| Ozone | | | | |
| nofire | 0.54 | **0.77** | 18.01 | -10.62 |
| f2no | 0.54 | 0.77 | 17.91 | -10.08 |
| f2all | **0.56** | 0.76 | **17.43** | **-8.46** |
| PM$_{2.5}$ | | | | |
| nofire | 0.37 | 0.39 | **5.02** | **2.65** |
| f2no | 0.39 | 0.44 | 5.42 | 3.01 |
| f2all | **0.39** | **0.44** | 5.48 | 3.07 |
| PM$_{10}$ | | | | |
| nofire | 0.25 | 0.54 | 8.19 | -2.65 |
| f2no | **0.31** | **0.57** | **8.08** | -2.26 |
| f2all | 0.29 | 0.53 | 9.08 | **-1.91** |
| AOD | | | | |
| nofire | **0.44** | 0.43 | 0.13 | -0.10 |
| f2no | 0.38 | **0.47** | 0.12 | -0.09 |
| f2all | 0.37 | 0.45 | **0.12** | **-0.09** |

**Table 2.** *Statistical scores for the surface ozone, PM$_{2.5}$, PM$_{10}$ ($\mu$g.m$^{-3}$) concentrations and AOD (a.d.) by comparison with EEA and AERONET measurements and the three simulations nofire, f2no and f2all. Scores are aggregated for all stations and the spatial correlation is added to the temporal correlation. Calculations are done over the entire modelled period (July and August 2022).*

values of surface ozone concentrations (as seen in Figure 10). For the two sites, the same behavior is observed: during July, the differences are between *nofire* and *f2all*: the *f2no* case is overlaid to the *f2all* case, meaning that the change is due to the addition of the biomass burning fluxes but not to impact of these fires on the surface. But during August, the differences change:

the simulations *nofire* and *f2no* are very close and the differences between *f2all* and *obs* are larger. It means that the impact on ozone is not due to active fires but to the impact of previous fires on the surface. During the month of August, the differences between *f2all* and *f2no* increase in time.

For each location, ozone surface concentrations display large differences between the simulation with no fires *nofire* and with the fires and the retroactions *f2all* as shown in Figure 12. Several periods are defined to see the time change of these

250 maxima values. Each period lasts two weeks: from 16 to 31 July, from 1 to 15 August and from 16 to 31 August 2022. For the three periods, the addition of fires induces an increase of surface ozone concentrations. In average over two weeks, this increase is ≈ 6 $\mu$g.m$^{-3}$ at the maximum. For the first period, the increase is mainly over continent, except the large plume coming for the Landes fires and going to the West, over the Atlantic sea. The second area of large additional concentrations is at the border of Portugal and Spain, due to Portuguese wildfires. During the second period, and due to several atmospheric

circulations, the additional ozone concentrations are modelled all over the domain. Positive differences have peaks again over



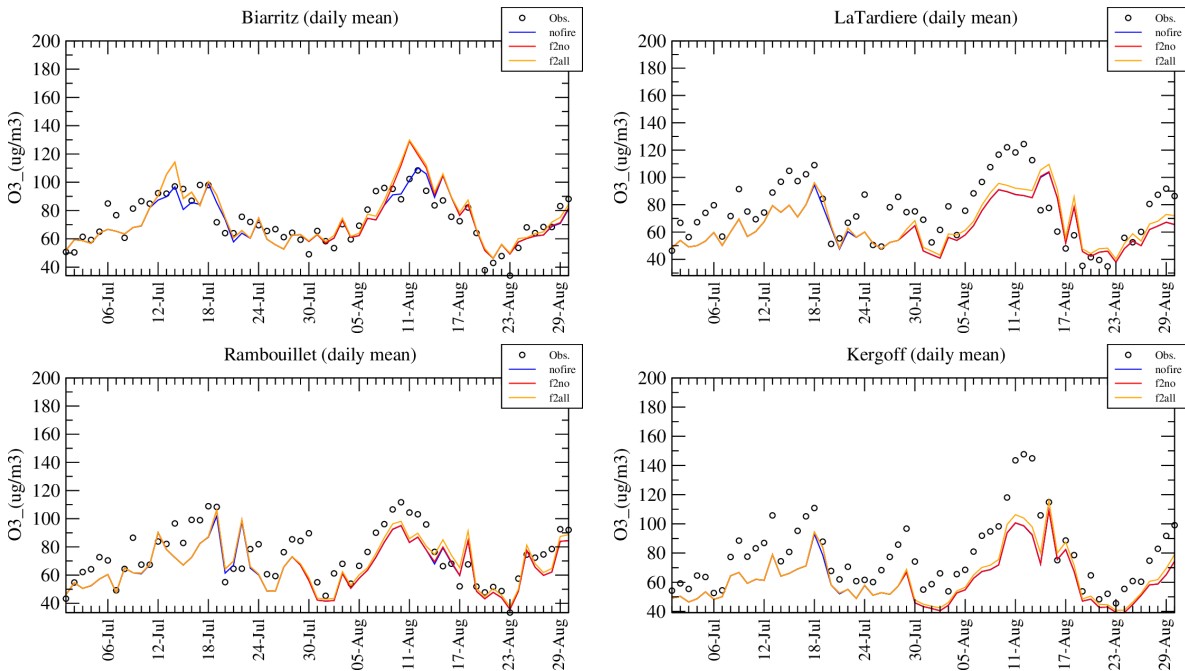

**Figure 10.** *Time-series of daily averaged surface ozone concentrations in Biarritz, LaTardiere, Rambouillet and Kergoff.*

Landes and Portugal, but also over the Pyrenean and Alps. In the North of France, additional ozone may reach 5 $\mu$g.m$^{-3}$, when it was only $\approx 1\mu$g.m$^{-3}$ during the July period. For the second period of two weeks in August, the ozone differences remain positive and are more located in the eastern part of the modelled domain, in Germany, Switzerland and Italy.

## 4.2 Observed and modelled exceedances

In order to quantify the impact of the fires as well as their impact on the surface, on the modelled ozone concentrations, Table 3 presents the number of exceedances of the daily maximum surface concentrations compared to thresholds. These exceedances are calculated station by station and two thresholds are selected: 120 and 180 $\mu$g.m$^{-3}$. These exceedances are independently counted for the observations and the three simulations: *nofire*, *f2no* and *f2all*. The first result is that there is much more exceedances with the observations than with the simulations. With the observations, all stations have at least one stations over

the daily maximum value of 120 $\mu$g.m$^{-3}$ during the two months (i.e 60 days). The stations with the most important number of observed exceedances are Diga (49), Gartringen and PuertoCotos (45) and OHP (42). For the threshold of 180 $\mu$g.m$^{-3}$ and for the observations, only a few stations are above this value: Kergoff (1), Brotonne (2), Fontainebleau (1), OHP (1), Vredepeel (1), Gartringen (2) and Diga (1).

With the model, the number of exceedances is always lower than with the observations. With the *nofire* simulation, there is a

270 non-negligible number of exceedances, showing that, obviously, the fires are not always responsible of ozone peaks in western Europe. For the threshold of 180 $\mu$g.m$^{-3}$, the model is able to catch only three exceedances, in Fontainebleau, Vredepeel



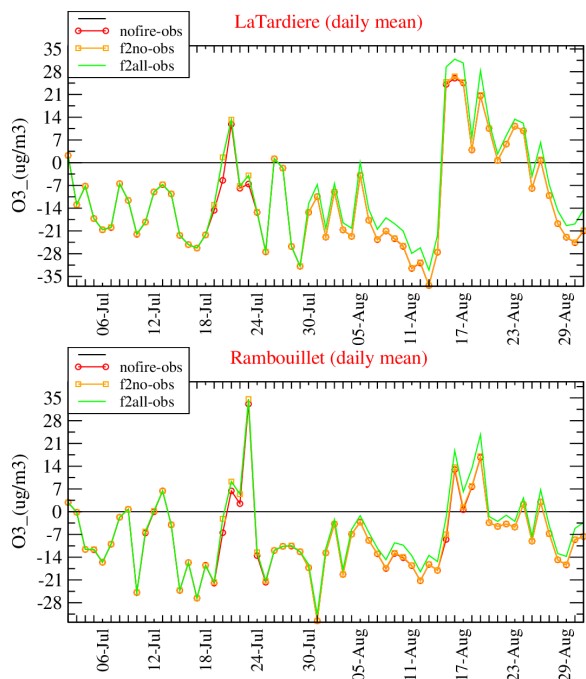

**Figure 11.** *Time-series of differences of surface ozone concentrations ($\mu$g.m$^{-3}$) in Airvault and Rambouillet.*

and Gartringen, when the observations showed exceedances for eight stations. For the simulation *f2no*, the stations where the addition fires causes a new exceedance are in green. There is 16 stations in this case and only for the threshold 120 $\mu$g.m$^{-3}$. But the increase in number of exceedance days is not very important: it is, for the most important part, one or two days more. For the simulation *f2all*, the values are in red when there is more days of exceedances compared to *f2no*. Almost all the stations are in this case: 25 stations (on 30) have more exceedances days than *f2no*, showing that the impact of the fires on the surface may have a non-negligible impact on surface ozone peaks. The additional number of exceedances are important: as an example, and for the threshold 120 $\mu$g.m$^{-3}$, the increase is from 4 to 9 in LaTardiere and StDenisAnjou, 10 to 14 in Tremblay, 22 to 27 in OHP. But, there is no change for the threshold 180 $\mu$g.m$^{-3}$: the number of exceedances remain the same and are lower than the observations. With these scores, it is noticeable that the addition of biomass burning emission fluxes has an impact on the daily maxima of surface ozone concentrations. This impact is only for the threshold 120 $\mu$g.m$^{-3}$ but not the one at 180 $\mu$g.m$^{-3}$. A second more important impact is when the retroaction of the fires on the surface is taken into account. Again, this is true for the threshold 120 $\mu$g.m$^{-3}$ but not for 180 $\mu$g.m$^{-3}$. In all cases, the modelled daily maxima remain lower than the observations.

## 5 Relative contributions of processes impacted by fires

Finally, this section presents an analysis of the processes involved in the impact of fires on the mineral dust and LAI. As presented in section 2.2, the fires emissions will have an impact at the surface by increasing the wind speed, the erodibility





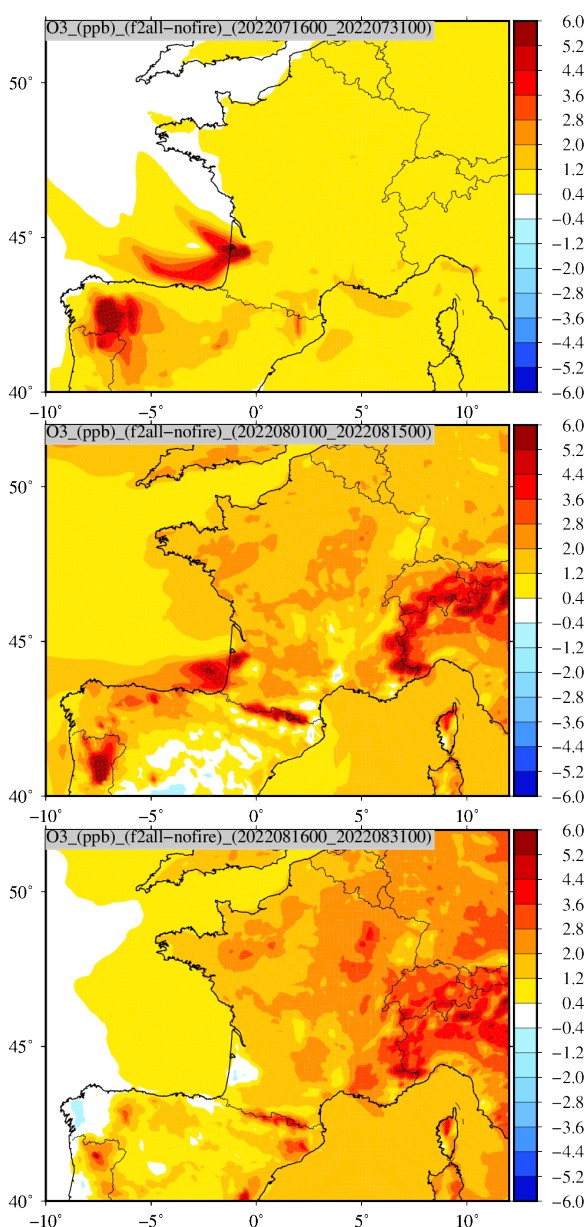

**Figure 12.** *Maps of differences of the maximal values of surface ozone concentrations (ppb), modelled in each model grid cell and for three consecutive two weeks periods.*

and decreasing the LAI. The decrease of LAI has an impact on biogenic emissions (less emissions) and dry deposition (less deposition). In Figure 10 and Figure 11, it has been shown that the differences between the simulation without (*nofire*) and with fires (*f2all*) may be divided into two distinct periods. First, during the month of July and when the fires were very active, there





| Location | obs | | *nofire* | | *f2no* | | *f2all* | |
|---|---|---|---|---|---|---|---|---|
| | 120 | 180 | 120 | 180 | 120 | 180 | 120 | 180 |
| Airvault | 12 | 0 | 4 | 0 | 4 | 0 | **6** | 0 |
| LaTardiere | 9 | 0 | 2 | 0 | **4** | 0 | **9** | 0 |
| Rambouillet | 19 | 0 | 9 | 0 | 9 | 0 | **10** | 0 |
| Peyrusse | 14 | 0 | 4 | 0 | 4 | 0 | **5** | 0 |
| Kergoff | 12 | 1 | 5 | 0 | **7** | 0 | 7 | 0 |
| StMalo | 6 | 0 | 6 | 0 | **7** | 0 | **9** | 0 |
| Mera | 14 | 0 | 5 | 0 | **6** | 0 | **7** | 0 |
| StDenisAnjou | 18 | 0 | 4 | 0 | **5** | 0 | **9** | 0 |
| Aytre | 9 | 0 | 4 | 0 | **5** | 0 | **7** | 0 |
| Zoodyss | 10 | 0 | 4 | 0 | **6** | 0 | **7** | 0 |
| Biarritz | 13 | 0 | 5 | 0 | **9** | 1 | **10** | 1 |
| Brotonne | 14 | 2 | 6 | 0 | 6 | 0 | **7** | 0 |
| Fontainebleau | 29 | 1 | 6 | 1 | 6 | 1 | **8** | 1 |
| Rageade | 20 | 0 | 3 | 0 | 3 | 0 | 3 | 0 |
| Verneuil | 11 | 0 | 3 | 0 | 3 | 0 | **7** | 0 |
| Tremblay | 19 | 0 | 10 | 0 | 10 | 0 | **14** | 0 |
| Vosges | 24 | 0 | 4 | 0 | 4 | 0 | **5** | 0 |
| OHP | 42 | 1 | 20 | 0 | **22** | 0 | **27** | 0 |
| Carling | 36 | 0 | 28 | 0 | 28 | 0 | **34** | 0 |
| MontsecOAM | 30 | 0 | 6 | 0 | **7** | 0 | 7 | 0 |
| Zorita | 10 | 0 | 8 | 0 | **9** | 0 | **10** | 0 |
| Valderas | 23 | 0 | 7 | 0 | **10** | 0 | **11** | 0 |
| PuertoCotos | 45 | 0 | 19 | 0 | **21** | 0 | 21 | 0 |
| Vredepeel | 12 | 1 | 14 | 2 | **15** | 2 | 15 | 2 |
| Moerkerke | 13 | 0 | 12 | 0 | **13** | 0 | **15** | 0 |
| Solling | 20 | 1 | 7 | 0 | 7 | 0 | **9** | 0 |
| Gartringen | 45 | 2 | 13 | 1 | 13 | 1 | **17** | 1 |
| Payerne | 27 | 0 | 18 | 0 | 18 | 0 | **22** | 0 |
| Diga | 49 | 1 | 0 | 0 | 0 | 0 | **3** | 0 |
| Hunsr | 28 | 0 | 11 | 0 | **12** | 0 | **16** | 0 |

**Table 3.** *Number of exceedances of daily maximum surface ozone concentrations recorded between 1st July and 31th August 2022 for the EEA stations and for the thresholds 120 and 180 $\mu g.m^{-3}$. Values are in green when the number of exceedances is different between nofire and f2no. Values are in red when the number of exceedances is different between f2no and f2all.*

is a direct impact of the fires on the ozone concentrations. Taking into account the retroaction has no impact, the differences between the simulations *f2all* and *f2no* being negligible. For the second part of the modelled period, in August, but this time the impact of fires is clearly highlighted with modelled differences observed between *f2all* and *f2no*. It means that the impact



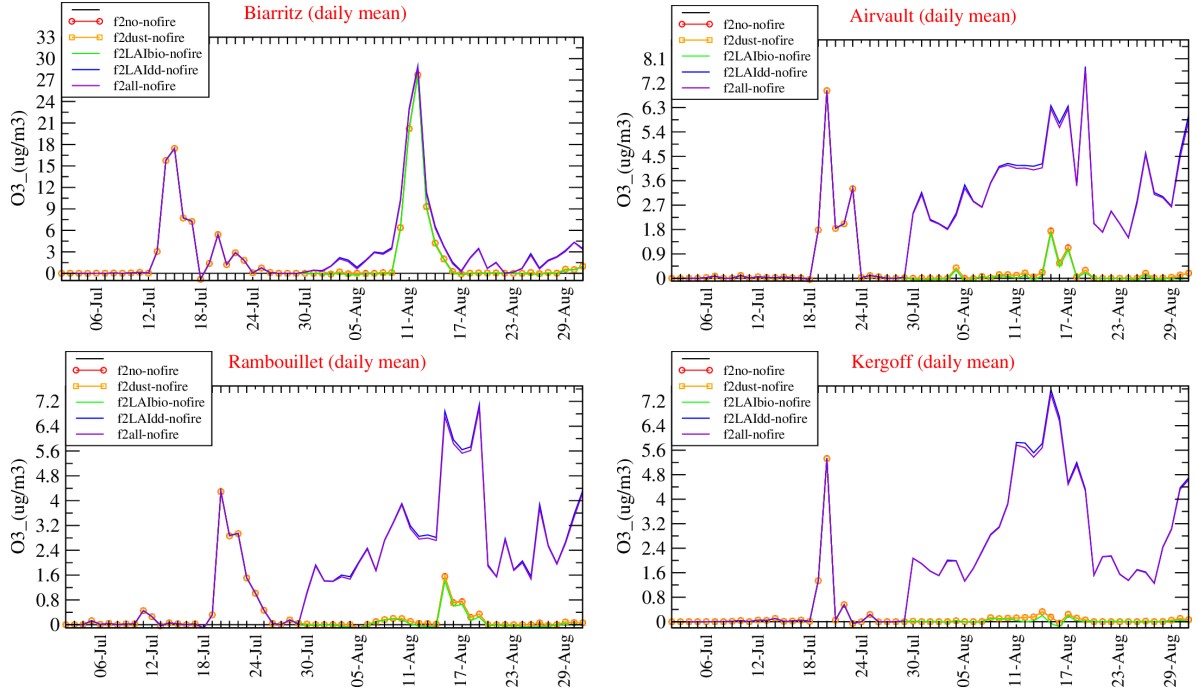

**Figure 13.** *Time-series of differences on ozone ($\mu g.m^{-3}$) surface concentrations in Biarritz, Airvault, Rambouillet and Kergoff. The differences are all model versus model and for all simulations with fires emissions against the simulation with no fires.*

of fires on ozone exists and is not due to a direct emission of pollutant but to a secondary effects of the fire on the surface then on ozone production.

The question is which process had the greatest effect on ozone production. Three additional simulations were performed and may be classified between *f2no* and *f2all*, as described in Table 1. Time series of differences between the simulations and *nofire* are presented in Figure 13 for surface ozone concentrations ($\mu g.m^{-3}$). Four locations are selected with Biarritz (south of France and the fires), Airvault (close to the Landes fires), Rambouillet (close to the Paris area) and Kergoff (Brittany). For the four sites, two peaks of differences are modelled. The first one in July is directly the impact of fires on ozone concentrations

and the second one in August is the indirect impact of the landuse change on the ozone production. For the first peak, the behavior is the same for all simulations: the difference between simulations with the fire and the simulation without the fires is the same for all configurations, meaning that the landuse changes has no impact during the fires or immediately after. The behavior is different for the second peak occurring in August. In Biarritz, the additional part of ozone added with the fires is important and reach 30 $\mu g.m^{-3}$. At the peak time, this impact is mostly due to the fires emissions directly. A small contribution

appears not due to the *f2dust* and *f2LAIbio* simulations, representing a few $\mu g.m^{-3}$. The behavior is different for the three other sites. The increase due to fires may reach 6 to 8 $\mu g.m^{-3}$, but this increase is mainly due to only one simulation, *f2LAIdd*. The other differences, due to *f2dust* and *f2LAIbio* remain with the same value that the simulation with no impact *f2no*. The direct



impact of the fires is only $\pm$ 1 $\mu$g.m$^{-3}$ for the sites. It means that after fires, one month later, the impact on vegetation leads to less dry deposition then much more concentration of ozone at the surface. The impact on mineral dust and biogenic emissions is not a first order impact for this pollution episode.

## 6 Conclusions

In this study, we simulated the summer 2022 with the CHIMERE model, forced by the ECMWF IFS meteorological fields, and over western Europe in order to model the huge fires events observed in the Landes forest. The model was able to simulate both ozone and PM$_{10}$ surface concentrations as well as the Aerosol Optical Depth during the two month of July and August 2022. Several simulations were performed, with and without fires, but also with and without impact of fires on the landuse, then the mineral dust emissions, the biogenic emissions and the dry deposition of gases.

Compared to observations, the implemetation of the Landes fires in the emissions improves the spatial and temporal correlation, the bias and the RMSE for almost all studied pollutants. With time-series in several locations in France, it has been shown that the model is able to retrieve the timing and the magnitude of the pollution peaks due to the fires. The simulations also showed that the Landes fires were not the only fires event during this summer and the results showed huge fires also in Spain and Portugal, transported to the North in the South of France. At the same time, mineral dust emissions from North Africa are also transported to southern France.

Calculations of ozone daily maxima and their comparison to threshold values (120 and 180 $\mu$g.m$^3$) showed that the fires are responsible of a lot of increase on ozone peaks during this period. But globally, the summer was not a very polluted summer, only a few stations showing surface concentrations above 180 $\mu$g.m$^3$ as daily maximum. The model underestimates the ozone peaks and no day above this threshold is modelled at any station. Taking into account the impact of fires on the landuse also changes the scores and increases the threshold exceedances and thus reduces the negative bias of the model on ozone peaks. It is therefore a process that should be considered in particular for the forecast of pollution in summer. More precisely, the most sensitive process for ozone is the fact that fires destroy vegetation and therefore reduce the LAI and therefore reduce the dry deposition of ozone and therefore increase its concentration in plumes downwind the fires. This process has an impact for a much longer period than fires, as the vegetation takes months or years to recover. The influence of the day-to-day surface state clearly shows the need for higher spatial and temporal frequency couplings between vegetation, surface and chemistry-transport models.

*Code availability.* The CHIMERE v2020 model is available on its dedicated web site https://www.lmd.polytechnique.fr and for download at https://doi.org/10.14768/8afd9058-909c-4827-94b8-69f05f7bb46d.



*Data availability.* All data used in this study, as well as the data required to run the simulations, are available on the CHIMERE web site download page https://doi.org/10.14768/8afd9058-909c-4827-94b8-69f05f7bb46d.

*Author contributions.* All authors contributed to the model development.

*Competing interests.* The authors declare that they have no conflict of interest.

*Acknowledgements.* We thank the investigators and staff who maintain and provide the AERONET data (https://aeronet.gsfc.nasa.gov/). European Environmental Agency (EEA) is acknowledged for their air quality station data that is provided and freely downloadable (https://www.eea.europa.eu/data-and-maps/data/aqereporting-8).

## Appendix A: Coordinates of measurements stations.

Table A1 and Table A2 present the coordinates and altitude above ground level of the stations for which the measurements are
345 used for the comparison with the model results.

## Appendix B: Maps of surface properties

Figure B1 presents the domain with 50 km resolution and the LAI database used before change by the fires and for the MEGAN biogenic emissions calculation.





| EEA stations | | | |
|---|---|---|---|
| **Station** | **Longitude** | **Latitude** | **Altitude** |
| **Name** | (°E) | (°N) | ASL (m) |
| Kergoff | -2.94 | 48.26 | 307. |
| StMalo | -2.00 | 48.65 | 5. |
| Mera | -0.45 | 48.64 | 309. |
| StDenisAnjou | -0.44 | 47.78 | 54. |
| Airvault | -0.13 | 46.82 | 100. |
| LaTardiere | -0.74 | 46.65 | 100. |
| Aytre | -1.11 | 46.13 | 10. |
| Zoodyss | -0.39 | 46.14 | 93. |
| Peyrusse | 0.17 | 43.62 | 230. |
| Biarritz | -1.55 | 43.47 | 70. |
| Brotonne | 0.75 | 49.49 | 10. |
| Fontainebleau | 2.64 | 48.35 | 127. |
| Rageade | 3.27 | 45.10 | 1040. |
| Verneuil | 2.61 | 46.81 | 182. |
| Rambouillet | 1.83 | 48.63 | 164. |
| Tremblay | 2.57 | 48.95 | 65. |
| Vosges | 7.12 | 48.49 | 770. |
| OHP | 5.71 | 43.93 | 668. |
| Carling | 6.76 | 43.43 | 5. |
| MontsecOAM | 0.72 | 42.05 | 1570. |
| Zorita | -0.16 | 40.73 | 619. |
| Valderas | -5.44 | 42.07 | 738. |
| PuertoCotos | -3.96 | 40.82 | 1200. |
| Vredepeel | 5.85 | 51.54 | 28. |
| Moerkerke | 3.36 | 51.25 | 3. |
| Solling | 9.55 | 51.70 | 295. |
| Gartringen | 8.90 | 48.64 | 466. |
| Payerne | 6.94 | 46.81 | 489. |
| Diga | 7.24 | 45.43 | 1576. |
| Hunsr | 7.19 | 49.74 | 650. |

**Table A1.** *List of the EEA sites used for the comparisons between measured and modelled surface concentrations.*



| AERONET stations | | | |
| --- | --- | --- | --- |
| **Station Name** | **Longitude** (°E) | **Latitude** (°N) | **Altitude** ASL (m) |
| Arcachon | -1.16 | 44.66 | 11. |
| Aubiere | 3.11 | 45.76 | 423. |
| Barcelona | 2.11 | 41.38 | 125. |
| Birkenes | 8.25 | 58.38 | 230. |
| Coruna | -8.42 | 43.36 | 67. |
| Evora | -7.91 | 38.56 | 293. |
| Kanzelhohe | 13.90 | 46.67 | 1526. |
| Lampedusa | 12.63 | 35.51 | 45. |
| Lille | 3.14 | 50.61 | 60. |
| Loftus | -0.86 | 54.56 | 159. |
| Madrid | -3.72 | 40.45 | 680. |
| Messina | 15.56 | 38.19 | 15. |
| Murcia | -1.17 | 38.00 | 69. |
| Napoli | 14.30 | 40.83 | 50. |
| Palma | 2.62 | 39.55 | 10. |
| Palaiseau | 2.20 | 48.70 | 156. |
| Paris | 2.33 | 48.86 | 50.0 |
| Saada | -8.15 | 31.62 | 420. |
| Saclay | 2.16 | 48.73 | 160. |
| Toulouse | 1.37 | 43.57 | 160. |
| Vienna | 16.33 | 48.23 | 266. |

**Table A2.** *List of the AERONET sites used for the comparisons between measured and modelled surface concentrations.*

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



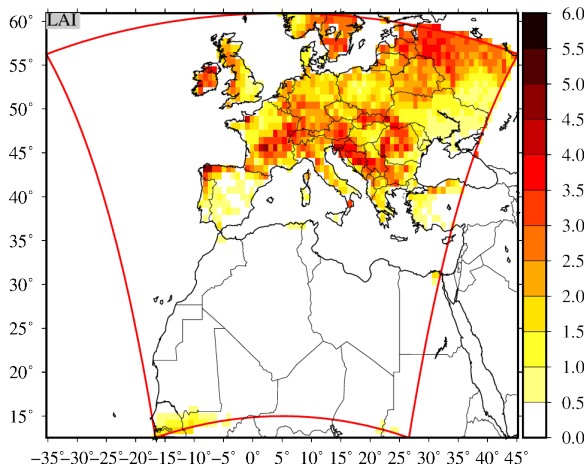

**Figure B1.** *Model domain ($\Delta x=50km$) with the Leaf Area Index (LAI in $m^2.m^{-2}$) used by the CHIMERE model and for the month of August.*

Grell, G., Freitas, S. R., Stuefer, M., and Fast, J.: Inclusion of biomass burning in WRF-Chem: impact of wildfires on weather forecasts, Atmospheric Chemistry and Physics, 11, 5289–5303, https://doi.org/10.5194/acp-11-5289-2011, 2011.

Guenther, A. B., Jiang, X., Heald, C. L., Sakulyanontvittaya, T., Duhl, T., Emmons, L. K., and Wang, X.: The Model of Emissions of Gases and Aerosols from Nature version 2.1 (MEGAN2.1): an extended and updated framework for modeling biogenic emissions, Geoscientific Model Development, 5, 1471–1492, https://doi.org/10.5194/gmd-5-1471-2012, 2012.

Haiden, T., Janousek, M., Vitart, F., Ben-Bouallegue, Z., Ferranti, L., Prates, F., and Richardson, D.: Evaluation of ECMWF forecasts, including the 2021 upgrade, https://doi.org/10.21957/xqnu5o3p, 2022.

Holben, B., Tanre, D., Smirnov, A., Eck, T. F., Slutsker, I., Abuhassan, N., Newcomb, W. W., Schafer, J., Chatenet, B., Lavenu, F., Kaufman, Y. J., Vande Castle, J., Setzer, A., Markham, B., Clark, D., Frouin, R., Halthore, R., Karnieli, A., O'Neill, N. T., Pietras, C., Pinker, R. T., Voss, K., and Zibordi, G.: An emerging ground-based aerosol climatology: Aerosol Optical Depth from AERONET, J. Geophys. Res., 106, 12 067–12 097, 2001.

Huang, Y., Wu, S., and Kaplan, J. O.: Sensitivity of global wildfire occurrences to various factors in the context of global change, Atmospheric Environment, 121, 86–92, https://doi.org/https://doi.org/10.1016/j.atmosenv.2015.06.002, interdisciplinary Research Aspects of Open Biomass Burning and its Impact on the Atmosphere, 2015.

Jaffe, D. A. and Wigder, N. L.: Ozone production from wildfires : a critical review, Atmospheric Environment, 51, 1–10, https://doi.org/10.1016/j.atmosenv.2011.11.063, 2012.

Kaiser, J. W., Heil, A., Andreae, M. O., Benedetti, A., Chubarova, N., Jones, L., Morcrette, J.-J., Razinger, M., Schultz, M. G., Suttie, M., and van der Werf, G. R.: Biomass burning emissions estimated with a global fire assimilation system based on observed fire radiative power, Biogeosciences, 9, 527–554, https://doi.org/10.5194/bg-9-527-2012, 2012.

Menut, L., C.Schmechtig, and B.Marticorena: Sensitivity of the sandblasting fluxes calculations to the soil size distribution accuracy, Journal of Atmospheric and Oceanic Technology, 22, 1875–1884, 2005.



Menut, L., Flamant, C., Turquety, S., Deroubaix, A., Chazette, P., and Meynadier, R.: Impact of biomass burning on pollutant surface concentrations in megacities of the Gulf of Guinea, Atmospheric Chemistry and Physics, 18, 2687–2707, https://doi.org/10.5194/acp-18-2687-2018, 2018.

Menut, L., Tuccella, P., Flamant, C., Deroubaix, A., and Gaetani, M.: The role of aerosol–radiation–cloud interactions in linking anthropogenic pollution over southern west Africa and dust emission over the Sahara, Atmospheric Chemistry and Physics, 19, 14 657–14 676,
https://doi.org/10.5194/acp-19-14657-2019, 2019.

Menut, L., Bessagnet, B., Siour, G., Mailler, S., Pennel, R., and Cholakian, A.: Impact of lockdown measures to combat Covid-19 on air quality over western Europe, Science of The Total Environment, 741, 140 426, https://doi.org/10.1016/j.scitotenv.2020.140426, 2020.

Menut, L., Bessagnet, B., Briant, R., Cholakian, A., Couvidat, F., Mailler, S., Pennel, R., Siour, G., Tuccella, P., Turquety, S., and Valari, M.: The CHIMERE v2020r1 online chemistry-transport model, Geoscientific Model Development, 14, 6781–6811,
https://doi.org/10.5194/gmd-14-6781-2021, 2021.

Menut, L., Siour, G., Bessagnet, B., Cholakian, A., Pennel, R., and Mailler, S.: Impact of Wildfires on Mineral Dust Emissions in Europe, Journal of Geophysical Research: Atmospheres, 127, e2022JD037 395, https://doi.org/https://doi.org/10.1029/2022JD037395, e2022JD037395 2022JD037395, 2022a.

Menut, L., Siour, G., Bessagnet, B., Cholakian, A., Pennel, R., and Mailler, S.: Impact of wildfires on mineral dust emissions in Europe,
Journal of Geophysical Research - Atmosphere, -, 1–27, 2022b.

Monahan, E. C.: In The Role of Air-Sea Exchange in Geochemical Cycling, chap. The ocean as a source of atmospheric particles, pp. 129–163, Kluwer Academic Publishers, Dordrecht, Holland, 1986.

Mora, O., Banos, V., Regolini, M., and Carnus, J.-M.: Using scenarios for forest adaptation to climate change: a foresight study of the Landes de Gascogne Forest 2050, Annals of Forest Science, 71, 313–324, https://doi.org/10.1007/s13595-013-0336-2, 2014.

Omar, A., Winker, D. M., Vaughan, M. A., Hu, Y., Trepte, C. R., Ferrare, R. A., Lee, K.-P., Hostetler, C. A., Kittaka, C., Rogers, R. R., Kuehn, R. E., and Liu, Z.: The CALIPSO Automated Aerosol Classification and Lidar Ratio Selection Algorithm, Journal of Atmospheric and Oceanic Technology, 26, 1994–2014, 2010.

Price, C. and Rind, D.: What determines the cloud-to-ground lightning fraction in thunderstorms?, Geophys. Res. Lett., 20, 463–466, https://doi.org/10.1029/93GL00226, 1993.

Rea, G., Turquety, S., Menut, L., Briant, R., Mailler, S., and Siour, G.: Source contributions to 2012 summertime aerosols in the Euro-Mediterranean region, Atmospheric Chemistry and Physics, 15, 8013–8036, https://doi.org/10.5194/acp-15-8013-2015, 2015.

Reid, J. S., Koppmann, R., Eck, T. F., and Eleuterio, D. P.: A review of biomass burning emissions part II: intensive physical properties of biomass burning particles, Atmospheric Chemistry and Physics, 5, 799–825, https://doi.org/10.5194/acp-5-799-2005, 2005.

San-Miguel-Ayanz, J., Durrant, T., Boca, R., Maianti, P., Liberta, G., Artés-Vivancos, T., Oom, D., Branco, A., de Rigo, D., Ferrari, D.,
Pfeiffer, H., Grecchi, R., and Nuijten, D.: Advance Report on Forest Fires in Europe, Middle East and North Africa 2021, Publications Office of the European Union, Luxembourg, 2022, ISBN 978-92-76-49633-5, JRC128678, 1–39, https://doi.org/10.2760/039729, 2022.

Sindelarova, K., Granier, C., Bouarar, I., Guenther, A., Tilmes, S., Stavrakou, T., Müller, J.-F., Kuhn, U., Stefani, P., and Knorr, W.: Global data set of biogenic VOC emissions calculated by the MEGAN model over the last 30 years, Atmospheric Chemistry and Physics, 14, 9317–9341, https://doi.org/10.5194/acp-14-9317-2014, 2014.

Tesche, M., Wandinger, U., Ansmann, A., Althausen, D., Muller, D., and Omar, A. H.: Ground-based validation of CALIPSO observations of dust and smoke in the Cape Verde region, Journal of Geophysical Research: Atmospheres, 118, 2889–2902, https://doi.org/10.1002/jgrd.50248, 2013.



Toreti, A., Bavera, D., Acosta Navarro, J., Cammalleri, C., de Jager, A., Di Ciollo, C., Hrast Essenfelder, A., Maetens, W., Magni, D., Masante, D., Mazzeschi, M., Niemeyer, S., and Spinoni, J.: Drought in Europe August 2022, Tech. Rep. JRC130493, Joint Research Center, https://doi.org/10.2760/264241, publications Office of the European Union, Luxembourg, 2022.

Vautard, R., B.Bessagnet, M.Chin, and Menut, L.: On the contribution of natural Aeolian sources to particulate matter concentrations in Europe: testing hypotheses with a modelling approach, Atmospheric Environment, 39, 3291–3303, 2005.

Vieira, D., Borrelli, P., Jahanianfard, D., Benali, A., Scarpa, S., and Panagos, P.: Wildfires in Europe: Burned soils require attention, Environmental Research, 217, 114 936, https://doi.org/https://doi.org/10.1016/j.envres.2022.114936, 2023.

Wagner, R., Jähn, M., and Schepanski, K.: Wildfires as a source of airborne mineral dust – revisiting a conceptual model using large-eddy simulation (LES), Atmospheric Chemistry and Physics, 18, 11 863–11 884, https://doi.org/10.5194/acp-18-11863-2018, 2018.

Wang, X., Zhang, L., and Moran, M. D.: Development of a new semi-empirical parameterization for below-cloud scavenging of size-resolved aerosol particles by both rain and snow, Geoscientific Model Development, 7, 799–819, https://doi.org/10.5194/gmd-7-799-2014, 2014.

Winker, D., Pelon, J., Coakley Jr., J. A., Ackerman, S. A., Charlson, R. J., Colarco, P. R., Flamant, P., Fu, Q., Hoff, R. M., Kittaka, C., Kubar, T. L., Le Treut, H., McCormick, M. P., Megie, G., Poole, L., Powell, K., Trepte, C., Vaughan, M. A., and Wielicki, B. A.: The CALIPSO Mission: A Global 3D View of Aerosols and Clouds, Bulletin of the American Meteorological Society, 91, 1211–1229, 2010.

Zhang, L., Gong, S., Padro, J., and Barrie, L.: A size-segregated particle dry deposition scheme for an atmospheric aerosol module, Atmospheric Environment, 35(3), 549–560, 2001.