# Peer review of "Impact of Landes forest fires on air quality in France during the 2022 summer."

_EGUsphere, 2023_

## Author Comment (AC1)

Answer to reviewers for the article
**egusphere-2023-421** to ACP
*"Impact of Landes forest fires on air quality in France during the summer 2022"*
by L.Menut, A.Cholakian, G.Siour, R.Lapère, R.Pennel, S.Mailler and B.Bessagnet.

**Answer to the Editor**

Dear Editor,

Thanks a lot for the reviews. We did not found major criticism from the reviewers (about the methodology or the results) but several needs for more explanations and references. Then, we added several sentences (reproduced below) to give more details about the used tools, the methodology and the results.
We made all proposed changes in the revised manuscript. Please note that answers are in blue and after each reviewer's remark.

Best regards,
Laurent Menut
May 19, 2023

**1  Reviewer #1**

**Reviewer #1 Evaluations:**

*Answer:*
We acknowledge the Reviewer #1 for the review, the proposed improvements and the positive conclusion.

Review of Menut et al. "Impact of Landes forest fires on air quality in France during the summer 2022"

**Comments:**

Menut et al. simulate the air quality impacts during a forest fire episode in France using the CHIMERE chemical transport model, also, uniquely, estimating the impacts of burned area on dust and biogenic emissions. Overall, this is a well-written, well-executed and focused case study with an interesting and fairly novel set of experiments. I have no major comments and only the following mostly editorial suggestions:

- Title: change "..in France during the summer 2022" to "...in France during the 2022 summer"

  *Answer:*
  The title is now: Impact of Landes forest fires on air quality in France during the 2022 summer.

- L20: suggest omitting "mechanically"

  *Answer:*
  Correction done.

- L47: change "ii) do the biomass burning..." to "ii) does the biomass burning..."

  *Answer:*
  Correction done.

- L49: change "responsible of" to "responsible for"

  *Answer:*
  Correction done.

- L63: omit "well" from "well capture"

  *Answer:*
  Correction done.

- L72: please provide a brief description of Kaiser et al.' (2012) based Global Fire Assimilation biomass burning product, mentioning that it is FRP-based, as distinct from the burned area, used to describe the 2022 fire episode

  *Answer:*
  It is corrected and the new sentence is: "The biomass burning emissions are those of CAMS as described in (Kaiser et al., 2012) and presented in Figure 1 for the modelled domain with 15 km resolution. Biomass burning fluxes are calculated at the global scale and with a system assimilating MODIS satellite observations of Fire Radiative Power (FRP). Burnt area are also provided but here used only for the scheme presented in this study, not for the fluxes calculation. At $0.5 \times 0.5$ horizontal resolution, these fluxes are projected on the CHIMERE grid."

[Figure]

Figure 1: *Time-averaged surface flux of CO emitted by fires during the months of July and August 2022 and calculated using the CAMS fires product. The studied Landes fires are those located around the longitude $0^o E$ and the latitude $45^o N$.*

- L72: please provide a brief description of the surface flux vertical distribution - are these FRP-based plume heights distributed as part of GFAS, or something else?

  *Answer:*
  The injection height and shape of the vertical profile are completely parameterized in CHIMERE. The following sentence was added: "The injection is height is parameterized following the Sofiev et al. (2012) scheme and the shape of the vertical injection is parameterized using the Veira et al. (2015) scheme."

- Figure 3: please point out the area of the Landes fires

*Answer:*
The following was added in the caption: "The studied Landes fires are those located around the longitude 0$^o$E and the latitude 45$^o$N."

- L88: change "responsible of less dry deposition" to "responsible for less dry deposition"

  *Answer:*
  Correction done.

- L93: change "have in common to cover the period from 15 June to..." to "have a common period of 15 June to..."

  *Answer:*
  Correction done.

- L95: here and at L290, I wasn't sure what was meant by 'retroactions'. Would 'interactions' be suitable?

  *Answer:*
  'retroaction' is correct because there is always action of meteorology on aerosol behaviour (transport, mixing, deposition). The fact to add the 'online coupling' add action of aerosol on meterology. Then , in this case, there is retroaction.

- L152: change "transported toward west" to "transported westward"

  *Answer:*
  Correction done.

- L161: this sentence was a bit hard to understand, but I think would start with "The limitions of the simulations are probably..."

  *Answer:*
  Yes, OK, the sentence was changed accordingly.

- L164: can omit "results" from "model concentrations results"

  *Answer:*
  OK

- L169: change "in altitude" to "at altitude"

  *Answer:*
  OK

- L173: change "It is meaning.." to "It means..."

  *Answer:*
  OK

- L216: change "Data are daily averaged" to "Data are averaged daily"

  *Answer:*
  OK

- L224: change "Values are not very high as daily mean" to "Values are not as high as the daily mean"

  *Answer:*
  OK

- L254: change "several atmospheric circulations" to "several synoptic events"?

  *Answer:*
  OK

- L262: please describe briefly how the two thresholds were selected

  *Answer:*
  These two tresholds are inspired by the real European thresholds of air quality management. In our case, the calculaiton of these 'alerts' is not exactly the same as for the regulation of air quality, since we are not calculating average over several hours but only the fact that, at least one hour per day, the concentrations are higher than a specific treshold.

- L209: change from "The impact on mineral dust..." to "The impact of mineral dust..."?

  *Answer:*
  It is really the impact of fires on mineral dust here. The sentence was changed to be more clear and is now: " The impact of fires on mineral dust and biogenic emissions is not a first order impact for this pollution episode."

- L319: change "able to retrieve" to "able to capture"

  *Answer:*
  OK done

- L324: by "globally", do you mean "regionally"?

  *Answer:*
  No, sorry, it is "overall" here. This was corrected.

**2 Reviewer #2**

**Reviewer #2 Evaluations:**

The study by Menut et al. explores the air pollution effects of intense wildfires that occur in the summer of 2022 in France, using an atmospheric chemical-transport model. A range of observational constraints are also utilized for the study, as well as sensitivity experiments with the model, which provide insights into the role of feedbacks via changes in the leaf area index and the dust emissions, along the direct effects of the wildfire emissions themselves. The manuscript is well written and well within the scope of the journal, while it provides insights that will be useful for the evolving field of fire-land-atmosphere interactions. There is one key concern that I have, and a few more minor ones. If those are addressed, I believed the manuscript will be suitable for publication.

**MAIN COMMENT:**

While the first part of the manuscript studies both aerosols and ozone, the part that utilizes the sensitivity simulations to study feedbacks via processes impacted by fires focuses only on ozone. Why is that? Surely the dust perturbation should have implications for PM, and possibly the other simulations too. Please explain and expand the analysis/discussion if/as needed.

*Answer:*
The choice to focus mostly on ozone was done for several reasons:

1. Because the impact of fires on dust has an impact on aerosols, but the impact on LAI then gas dry deposition and gaseous biogenic emissions will have mostly an impact on gaseous species but a negligible impact on aerosols.

2. A detailed impact of fires on mineral dust was already and extensively studied in Menut et al. (2022), and we tried to be as new as possible.

3. ... and finally to not have a too long paper.

To explain that more clearly, the following sentence was added at the beginning of the section 4:
*"In this section, the impact of fires on surface ozone concentrations is quantified. Details about the sensitivity simulations are also presented, including all simulations. Results are presented for ozone only because the impact of fires on LAI then biogenic emissions and dry deposition impacts gaseous species but has a negligible impact on aerosols."*

Also, via what mechanism does dust impact ozone in the model? Since chemicals and aerosols are not allowed to influence radiation/meteorology, it is probably implied that this is due to photolysis? This should be more clearly explained and discussed.

*Answer:*
The possible impact of mineral dust on ozone is via the photolysis rates only. Of course, if the "online" coupling is activated, with direct and indirect effects on aerosols on clouds and radiation, the impact on ozone would increase due to these additional processes.

**MINOR COMMENTS:**

- Abstract: The abstract should refer more to the studies results and less to its hypotheses.

  *Answer:*
  Following this remark, the abstract was simplified and shortened. But the novelty of this study is to add these new processes, thus it seems important to cite them.

- Page 2, Line 28: What does "still" refer to?

  *Answer:*
  It means that at the surface, the fire seems to be finished. But it is not really the case and this explains why the fire resumed its activity in the same area a few weeks later. The new sentence is now: "However, the Landiras fire, apparently finished at the surface but still propagating underground due to the presence of peat, began active again on 9 August, burning another..."

- Page 3, Line 66: "Several tens of chemical species, gas and aerosol, are modelled": more detail needed. Also, how is photolysis treated?

  *Answer:*
  As the goal of this study is not to detail too much the model used, already described in the release versions papers, some references were simply added here. The following text was added: "For gases, the MELCHIOR 2 scheme is used as described in Menut et al. (2013) and Mailler et al. (2017)."

- Page 3, Line 72: Given that biomass burning emissions are central to the study, there should be a bit more insight into how CAMS emissions are derived.

  *Answer:*
  It is also a remark from Reviewer #1 and details were added in the manuscript.

- Page 8, Line 140: correspond -> corresponds

    *Answer:*
    OK corrected.

- Page 8, Lines 140-141: "The model overestimates the measurements but is composed of Primary Organic Matter (POM), signature of the biomass burning" - what does this mean? What does the second half of the sentence tell us?

    *Answer:*
    OK, it is not clear. The sentence was rewritten as: "The model overestimates the measurements and is composed of Primary Organic Matter (POM). The predominance of this species in the aerosol composition is a signature of the biomass burning."

- Figure 5: Three panels are described in the caption, but four are shown. Also. The maps depict the differences, which needs to be mentioned in the caption.

    *Answer:*
    Yes, right. The caption is corrected and is now: "Maps of surface concentrations of $PM_{10}$ ($\mu$g.m$^{-3}$) for the 18 July 2022 at 12:00 UTC, 19 July 2022 at 12:00 UTC and 20 July 2022 at 00:00 and 12:00 UTC."

- Page 9, Line 160: "Finally, the impact of fires induces positive differences only" - perhaps add a second part to this sentence, to show why this is not entirely obvious (e.g. "..., indicating that the negative feedbacks from fire emissions do not outweigh the effects of direct emissions and positive feedbacks at any location.")

    *Answer:*
    Thanks for the suggestion, it was added accordingly.

- Page 9, Lines 161-162: Why is this a drawback? (also, "lack" may not be the most suitable word to use here)

    *Answer:*
    Yes, "lack" was changed to "limitations". It is not really a drawback and the sentence was removed here. The presentation of the vertical strcuture is here too early. The discussion already exist later in the subsection "Vertical transport..." with the sentence: "The differences between the time-series of AOD and surface concentrations of $PM_{10}$ show that the fire plume might have been transported aloft without high concentrations being present at the surface. To verify this hypothesis with the simulations, vertical sections are presented".

- Page 10. Lines 173-174: Any ideas why?

    *Answer:*
    Yes, and this is exactly what is being discussed in the "vertical cross-section" subsection. The plume is too much in altitude (due to vertical mixing) and is not enough transported close to the surface.

- Page 10, Lines 181-184: The EES and what it is should be explained more clearly.

    *Answer:*
    Yes, OK. The following lines are added:
    In this Figure, note that the Efficient Extinction Section (EES and noted $\sigma_p^{\mathrm{ext}}(z, \lambda)$) coefficient is superimposed (in dashed line). This coefficient is used to the AOD calculation, $\tau_{\mathrm{ext}}(\lambda, z)$, for one atmospheric layer depth $\Delta z$ and one specific wavelength $\lambda$, (Stromatas et al., 2012), such as:

$$\tau_{\text{ext}}(\lambda, z) = \int_{\Delta z} \sigma_p^{\text{ext}}(\lambda, z')\mathrm{d}z' \tag{1}$$

with the extinction coefficient (by particles), $\sigma_p^{\text{ext}}(z, \lambda)$ $(m^{-1})$ as:

$$\sigma_p^{\text{ext}}(z, \lambda) = \int_{R_{\min}}^{R_{\max}} \pi R^2 Q_{\text{ext}}(\eta, R, \lambda) \cdot N_p(R, z)\mathrm{d}R \tag{2}$$

where $Q_{\text{ext}}$ is the extinction efficiency, depending on the refractive index $(\eta)$, the particles radius $(R)$, the wavelength $(\lambda)$, and $N_p$ is the particle concentration in number $(m^{-3})$. For the Figure, the EES is normalized to have the same order of magnitude than the maximum of concentration. It appears that its maximum corresponds to a minimum of concentration in the size distribution: it means that the AOD calculation is very sensitive to the size distribution and the number of bins of the model (even if here it is concentrations at the surface only).

- Page 10, Lines 185-186: Yes, but there is also quite a large difference in PPM.

  *Answer:*
  Yes, of course, this was added in the text.

- Line 227: others simulation -> other simulations

  *Answer:*
  OK corrected.

- Figures 10 and 11: From these figures, the effect of fires seems to basically be minimal, if not negligible. Is this not the main conclusion of this section? Because there is plenty of analysis in this sub-section, but this bottom-line conclusion is not really coming across clearly.

  *Answer:*
  Yes, we agree and it is already noted at the beginning of the section, line 219: *"For the four stations presented in Figure 10, Biarritz, LaTardiere, Rambouillet and Kergoff, located at various ranges from the fires (LaTardiere being the closest one), there is no important impact of the fires emissions on daily mean surface ozone concentrations. The concentrations vary a lot from one week to another, but the simulated concentrations are very close to each other."*

- Page 19, Line 305: "appears not" maybe should be "appears"?

  *Answer:*
  yes, thanks and of course, corrected. The new sentence is: "A small contribution of a few $\mu$g.m$^{-3}$ is diagnosed with the *f2dust* and *f2LAIbio* simulations."

- Page 20, Line 324: "globally" can be a misleading use of word here.

  *Answer:*
  Yes, right, same remark as the Reviewer#1 and a much more correct word is "overall".

**References**

Kaiser, J. W., Heil, A., Andreae, M. O., Benedetti, A., Chubarova, N., Jones, L., Morcrette, J.-J., Razinger, M., Schultz, M. G., Suttie, M., and van der Werf, G. R.: Biomass burning emissions estimated with a global fire assimilation system based on observed fire radiative power, Biogeosciences, 9, 527–554, doi: 10.5194/bg-9-527-2012, 2012.

Mailler, S., Menut, L., Khvorostyanov, D., Valari, M., Couvidat, F., Siour, G., Turquety, S., Briant, R., Tuccella, P., Bessagnet, B., Colette, A., Létinois, L., Markakis, K., and Meleux, F.: CHIMERE-2017: from urban to hemispheric chemistry-transport modeling, Geoscientific Model Development, 10, 2397–2423, doi: 10.5194/gmd-10-2397-2017, 2017.

Menut, L., Bessagnet, B., Khvorostyanov, D., Beekmann, M., Blond, N., Colette, A., Coll, I., Curci, G., Foret, F., Hodzic, A., Mailler, S., Meleux, F., Monge, J., Pison, I., Siour, G., Turquety, S., Valari, M., Vautard, R., and Vivanco, M.: CHIMERE 2013: a model for regional atmospheric composition modelling, Geoscientific Model Development, 6, 981–1028, doi: 10.5194/gmd-6-981-2013, 2013.

Menut, L., Siour, G., Bessagnet, B., Cholakian, A., Pennel, R., and Mailler, S.: Impact of Wildfires on Mineral Dust Emissions in Europe, Journal of Geophysical Research: Atmospheres, 127, e2022JD037395, doi: https://doi.org/10.1029/2022JD037395, e2022JD037395 2022JD037395, 2022.

Sofiev, M., Ermakova, T., and Vankevich, R.: Evaluation of the smoke-injection height from wild-land fires using remote-sensing data, Atmospheric Chemistry and Physics, 12, 1995–2006, doi: 10.5194/acp-12-1995-2012, 2012.

Stromatas, S., Turquety, S., Menut, L., Chepfer, H., Cesana, G., Pere, J., and Bessagnet, B.: Lidar signal simulation for the evaluation of aerosols in chemistry-transport models, Geoscientific Model Development, 5, 2012.

Veira, A., Kloster, S., Wilkenskjeld, S., and Remy, S.: Fire emission heights in the climate system - Part 1: Global plume height patterns simulated by ECHAM6-HAM2, Atmospheric Chemistry and Physics, 15, 7155–7171, doi: 10.5194/acp-15-7155-2015, 2015.